# House design and risk of malaria, acute respiratory infection and gastrointestinal illness in Uganda: A cohort study

**Alex K. Musiime**[1,2]*, **Paul J. Krezanoski**[3], **David L. Smith**[4], **Maxwell Kilama**[1], **Melissa D. Conrad**[3], **Geoffrey Otto**[1], **Patrick Kyagamba**[1], **Jackson Asiimwe**[1], **John Rek**[1], **Joaniter I. Nankabirwa**[1,5], **Emmanuel Arinaitwe**[1], **Anne M. Akol**[2], **Moses R. Kamya**[1,5], **Sarah G. Staedke**[6], **Chris Drakeley**[7], **Teun Bousema**[8], **Steve W. Lindsay**[9], **Grant Dorsey**[3], **Lucy S. Tusting**[10,11]

1 Infectious Diseases Research Collaboration, Kampala, Uganda, 2 Department of Zoology, Entomology and Fisheries Sciences, College of Natural Sciences, Makerere University, Kampala, Uganda, 3 Department of Medicine, San Francisco General Hospital, University of California, San Francisco, United States of America, 4 Department of Health Metrics Sciences, University of Washington, Seattle, United States of America, 5 Department of Medicine, Makerere University College of Health Sciences, Kampala, Uganda, 6 Faculty of Infectious and Tropical Diseases, Clinical Research Department, London School of Hygiene & Tropical Medicine, London, United Kingdom, 7 Faculty of Infectious and Tropical Diseases, Department of Infection Biology, London School of Hygiene & Tropical Medicine, London, United Kingdom, 8 Department of Medical Microbiology, Radboud University Nijmegen Medical Centre, Nijmegen, The Netherlands, 9 Department of Biosciences, Durham University, Durham, United Kingdom, 10 Faculty of Infectious and Tropical Diseases, Department of Disease Control, London School of Hygiene & Tropical Medicine, London, United Kingdom, 11 Centre on Climate Change and Planetary Health, London School of Hygiene & Tropical Medicine, London, United Kingdom

* alexmedicare@yahoo.com

**Data Availability Statement:** The datasets used for this study are available publicly from the open-access clinical epidemiology database resource,

## Abstract

House construction is rapidly modernizing across Africa but the potential benefits for human health are poorly understood. We hypothesised that improvements to housing would be associated with reductions in malaria, acute respiratory infection (ARI) and gastrointestinal illness in an area of low malaria endemicity in Uganda. Data were analysed from a cohort study of male and female child and adult residents (n = 531) of 80 randomly-selected households in Nagongera sub-county, followed for 24 months (October 4, 2017 to October 31, 2019). Houses were classified as modern (brick walls, metal roof and closed eaves) or traditional (all other homes). Light trap collections of mosquitoes were done every two weeks in all sleeping rooms. Every four weeks, we measured malaria infection (using microscopy and qPCR to detect malaria parasites), incidence of malaria, ARI and gastrointestinal illness. We collected 15,780 adult female *Anopheles* over 7,631 nights. We collected 13,277 blood samples of which 10.2% (1,347) were positive for malaria parasites. Over 958 person years we diagnosed 38 episodes of uncomplicated malaria (incidence 0.04 episodes per person-year at risk), 2,553 episodes of ARI (incidence 2.7 episodes per person-year) and 387 episodes of gastrointestinal illness (incidence 0.4 episodes per person-year). Modern houses were associated with a 53% lower human biting rate compared to traditional houses (adjusted incidence rate ratio [aIRR] 0.47, 95% confidence interval [CI] 0.32–0.67, p<0.001) and a 24% lower incidence of gastrointestinal illness (aIRR 0.76, 95% CI 0.59–0.98, p =

ClinEpiDB, at https://clinepidb.org/ce/app/record/dataset/DS_51b40fe2e2.

**Funding:** This research report is supported by the National Institute of Allergy and Infectious Diseases (NIAID) as part of the International Centers of Excellence in Malaria Research (ICEMR) program (U19AI089674), an NIAID Career Mentored Award to PJK (K23AI139364) and the Fogarty International Center of the National Institutes of Health under Award Number D43TW010526. The content is solely the responsibility of the authors and does not necessarily represent the official views of the National Institutes of Health. SWL is supported by the Global Challenges Research Fund for Networks in Vector Borne Disease Research which is co-funded by BBSRC, MRC and NERC (BB/R00532X/1); JIN is supported by the Fogarty International Center (Emerging Global Leader Award grant number K43TW010365). LST is a Skills Development Fellow (#N011570) jointly funded by the UK Medical Research Council (MRC) and the UK Department for International Development under the MRC/DFID Concordat agreement. The funders had no role in study design, data collection and analysis, decision to publish, or preparation of the manuscript.

**Competing interests:** The authors declare that they have no competing interests.

0.04) but no changes in malaria prevalence, malaria incidence nor ARI incidence. House improvements may reduce mosquito-biting rates and gastrointestinal illness among children and adults. For the health sector to leverage Africa's housing modernization, research is urgently needed to identify the healthiest house designs and to assess their effectiveness across a range of epidemiological settings in sub-Saharan Africa.

## Introduction

Despite substantial improvements in health in sub-Saharan Africa (SSA) since 2000, mortality among children aged under five years remains high, with at least 34 countries unlikely to meet the Sustainable Development Goal 3 target of <25 deaths per 1,000 live births by 2030 [1]. Pneumonia, diarrhoea and malaria persist as the leading causes of mortality beyond the neonatal period, contributing 40% of the 2.7 million total deaths in children aged <5 years old across SSA in 2019 [2]. As SSA undergoes rapid urbanization, population growth and environmental changes, there is an urgent need to better understand the underlying social and environmental determinants of diseases that contribute significantly to morbidity and mortality [3].

Access to adequate housing, a human right and a core aim of Sustainable Development Goal 11 [4], is a promising yet neglected topic in SSA [5]. Features of inadequate housing, including indoor air pollution, overcrowding, unsafe water and poor sanitation, have long been recognised as risk factors for diarrhoea, respiratory disease and malaria, and provision of good-quality housing was a core pillar of early public health programs [6]. Today, improvements to housing are highly relevant to health in SSA. A recent analysis of 824,694 African children aged 0–5 years included in 77 nationally representative surveys between 2001 and 2017 found an 8%–18% lower odds of malaria, anaemia, undernutrition and diarrhoea among residents of modern houses compared to those living in traditional houses, adjusting for household wealth and other factors [7]. In most cases, housing with durable or finished walls and roofs, adequate sanitation, safe drinking water and sufficient living area per person should be more conducive to health than housing with rudimentary construction, inadequate water and sanitation and overcrowding [7]. For example, the primary African malaria vector, *Anopheles gambiae*, typically bites indoors at nighttime, so simple features such as screened windows and doors, metal roofs and closed eaves can reduce exposure to malaria infection [8].

Today, housing in SSA is rapidly transforming alongside economic development and population growth, with major potential benefits for health. Traditional houses constructed with thatch roofs, mud walls and open eaves are being widely replaced by houses with metal and tiled roofs and concrete and brick walls, and "improved" housing (with improved water and sanitation, sufficient living space and construction with durable materials) doubled from 11% in 2000 to 23% in 2015 in SSA [9]. These changes are an opportunity for better health. For example, housing improvements could avert an estimated 19 to 30 million of 213 million total annual malaria cases in SSA [8]. These improvements, however, cannot be fully leveraged without an understanding of effects across different health outcomes [3]. For example, sealing all openings in a house will keep out malaria vectors but will reduce air flow, potentially raising indoor temperature and discouraging use of insecticide-treated nets (ITNs) [10]. Conversely, improving ventilation by installing at least two large-screened windows on opposite walls reduces carbon dioxide ($CO_2$) concentration indoors and subsequent attractiveness of homes to mosquitoes [11]. Modern housing may offer protection against acute respiratory infection (ARI) due to a reduction in mould and better ventilation, although ventilation may be

impeded by ITN use [12]. Diarrhoea may decline in modern houses with cement floors that can be easily cleaned and, potentially, fewer flies if the house is well screened [13]. Overall, evidence on these complex links between house design and health remains insufficient to inform housing policies [10].

In Uganda, housing improvements have been linked with lower malaria infection and indoor mosquito density [14–16], but it is unknown how these improvements impact on other health outcomes. Here, to our knowledge, we present the first cohort study to investigate the association between modern housing and three major causes of child mortality in SSA: malaria, ARI and gastrointestinal illness. The study was conducted in rural Uganda following the reduction of malaria transmission to very low levels after mass ITN and indoor residual spraying (IRS) campaigns [17]. We tested the hypothesis that modern housing is associated with a lower incidence of malaria, ARI and gastrointestinal illness, compared to traditional housing.

## Materials and methods

### Study site

This analysis uses data from a cohort study of children and adults in rural Nagongera sub-county (00˚46'10.6"N, 34˚01'34.1"E), Tororo district, eastern Uganda [17]. The cohort study was designed to investigate optimal strategies for malaria control in Uganda using detailed longitudinal studies, with the sample size calculated according to this objective. The area is characterized by savannah grassland interspersed with rocky outcrops and wetlands. There are typically two rainy seasons each year, although during the study period there was a single main peak in May-June and a smaller peak in September-October that occurred earlier than expected [17]. Arable farming is an important livelihood. In Tororo, houses built with metal roofs and brick or concrete walls are becoming predominant, replacing thatch roofs and mud walls in line with the national trend; in Uganda, 67% of houses had a metal or tiled roof in 2016 compared to 55% in 2006 and 37% of houses had a finished brick or concrete wall in 2016 compared to 23% in 2006 [18,19]. For malaria, artemether-lumefantrine was adopted as the first-line treatment in 2006 in Tororo district, provided free of charge at public health facilities. Before vector control was implemented, Tororo had intense malaria transmission with an estimated *Plasmodium falciparum* annual entomological inoculation rate (EIR) of 562 infectious bites per person in 2001 and 125 in 2011–2012 [20]. Two rounds of universal free ITN distribution were conducted in November 2013 and June 2017, achieving approximately 80% population coverage [21]. Three rounds of IRS with bendiocarb (December 2014-January 2015, June-July 2015, and November-December 2015) and four rounds of IRS with Actellic (pirimiphos-methyl) (June-July 2016, July-August 2017, June-July 2018 and March-April 2019) were done. The implementation of IRS was associated with a 98% reduction in age-adjusted malaria incidence, which declined from 4.03 episodes per person-year before IRS to 0.10 episodes per person-year after the fifth and sixth rounds of IRS [22]. Nationwide prevalence of reported diarrhoea in the past two weeks was 21% and prevalence of reported cough with short and rapid breathing in the past two weeks was 15% in children aged 0–5 years in 2017 [18]. In Tororo, incidence of ARI has been observed to be relatively high compared to other districts of Uganda; during 2013–2016 the estimated mean annual pneumonia hospitalisation rate was 647 per 100,000 people [23].

### Enrollment and follow up of study participants

The present study was a continuation of a cohort study established in 2011 [24]. In October 2017, households still enrolled in the prior cohort study, plus new households randomly

selected from an enumeration list in Nagongera sub-county, were screened and enrolled if they met the following criteria: 1) at least two household members <5 years of age (or <10 years of age if enrolled in the prior cohort study), 2) no more than seven permanent residents (or nine permanent residents if enrolled in the prior cohort study), 3) no plans for the household to move from Nagongera sub-county in the next two years, and 4) the household head provided informed written consent to participate in the study. All permanent residents of enrolled households, regardless of age, were screened and enrolled in the cohort study if they met the following criteria: 1) the selected household was their primary residence, 2) residents agreed to come to the study clinic for any febrile illness, 3) residents agreed to avoid antimalarial medications outside the study and 4) provision of written informed consent. The cohort was dynamic such that during the study any permanent residents joining the household were screened for enrollment. Participants were followed for 24 months (to October 2019). At enrolment households were given sufficient ITNs (PermaNet; Vestergaard Frandsen, Denmark) to cover all household residents and replacement ITNs were provided free of charge at any time on request. Households were encouraged to attend a dedicated study clinic open seven days a week for all their medical care. Routine clinic visits were conducted every four weeks.

## Household characteristics

Each household was visited at baseline by trained study staff and a questionnaire administered to the head of the household to record the number of household occupants. Features of the house including main materials of the wall, roof, floor, number of sleeping rooms, type of eaves and number of windows were directly observed and recorded by trained study staff.

## Measurement of malaria, ARI and gastrointestinal illness

Participants with fever (tympanic temperature $\geq$ 38.0˚C) or reported history of fever in the previous 24 hours at the time of a clinic visit had a thick blood smear read immediately [25]. If the thick blood smear was positive, the patient was diagnosed with malaria, treated with artemether-lumefantrine and managed according to national guidelines. Routine clinic visits were conducted every four weeks, when a blood smear and a filter-paper sample were collected to measure microscopic and sub-microscopic malaria infections. Quantitative PCR (qPCR) was performed at the time of enrollment and at each routine visit to detect submicroscopic infections. DNA was extracted using Qiagen spin columns and extraction products were tested for the presence and quantity of *P. falciparum* DNA via a highly sensitive qPCR assay targeting the multicopy conserved *var* gene acidic terminal sequence, with a lower limit of detection of 1 parasite/mL [26]. ARI and gastrointestinal illnesses were diagnosed by passive case detection on reported symptoms either at a routine or spontaneous visit to the study clinic using the diagnostic criteria in S1 Table.

## Entomology

Entomological surveillance was conducted in all houses enrolled in the cohort study. From November 2, 2017 to October 20, 2019, indoor host-seeking mosquitoes were collected fortnightly in each house using miniature Centers for Disease Control light traps (Model 512; John W. Hock Company, Gainesville, Florida, USA). Light traps were positioned in each occupied bedroom with the light bulb 1m above the floor at the foot end of the bed where a person slept under an ITN. Traps were set at 19.00 h and collected at 07.00 h the following morning. The time that the room occupant went to bed the previous night was reported to study staff and recorded. If the occupant did not spend the night in the selected room or if

the trap was faulty, the data were excluded from the analysis. Specimens were maintained and sealed in collection cups and immediately delivered to the laboratory at Nagongera Health Centre. Individual mosquitoes were put in labeled micro-tubes and stored in ziplock bags with a desiccant. *Anopheles* mosquitoes were identified using standard keys [27] and PCR used to determine the species of mosquitoes belonging to the *An. gambiae s.l.* complex [20]. Infectivity of *Anopheles* with *P. falciparum* sporozoites was determined by enzyme-linked immunosorbent assay (ELISA) [20]. The head and thorax of each stored mosquito was separated from the rest of the body parts and ground in blocking buffer containing IGE-PAL CA-630. An aliquot of 50 μl was transferred to plates coated with monoclonal antibodies and positive and negative controls added [28]. After a series of incubation and washing, the plate was read at 405–414 nm using an ELISA plate reader to differentiate between infected and non-infected mosquitoes.

**Meteorological data.**   Monthly average rainfall data for the study period were estimated for Nagongera sub-county Tororo district using the National Aeronautics and Space Administration Tropical Rainfall Measuring Mission [29].

## Statistical analysis

House type was defined using a classification previously used in the study area [15,16]. Houses were classified as "modern" if they had all the following three features: a metal roof, brick wall and closed eaves. All other houses, typically with thatched roofs, mud walls, and open eaves, were classified as "traditional". House type was categorized both at the levels of the individual house and the individual sleeping room. Sanitation facility and drinking water source was defined as improved or unimproved using WHO Joint Monitoring Programme (WHO-JMP) criteria [30], which consider whether or not a sanitation facility adequately separates human excreta from human contact (e.g. improved toilets include latrines with washable slabs) and whether or not a drinking water source has adequate protection from outside contamination (e.g. improved sources include piped water, protected wells and rainwater). For the wealth index, principal component analysis was used to create a wealth index from nine variables: ownership of mobile telephones, radios, clocks, cupboards, sofas, and tables; access to improved sanitation; access to improved drinking water; and main mode of transport to the health facility. Households were ranked by wealth scores and grouped into tertiles to give a categorical measure of socioeconomic position as least poor, middle and poorest. Cross tabulations and Pearson's $\chi^2$ test were used to explore the relationship between house design and other household characteristics.

For each risk factor, we modelled its association with anopheline house density, a proxy for potential human biting rate (HBR; the number of adult female *Anopheles* caught per house per collection night), prevalence of microscopic and sub-microscopic malaria infection (measured by microscopy and qPCR) and the incidence of malaria, ARI and gastrointestinal illness (the number of new episodes of malaria, ARI or gastrointestinal illness per person years of observation). Negative binomial regression was used to model the number of adult female *Anopheles* caught per room and the number of malaria, ARI and gastrointestinal illness episodes per person, with the number of collection nights and person days included as an offset term in the model. The odds of parasitaemia at the time of each clinic visit were modelled using logistic regression. We adjusted for prespecified covariables including age, gender, household wealth, use of malaria interventions and quality of water and sanitation facilities. For all outcomes, robust standard errors were used to adjust for repeat measures (clustering) at the household level. Data analyses were done using Stata Version 16 (StataCorp, Texas) and R version 4.1.0 (R Core Team, Vienna).

### Ethics approval and consent to participate

Participants received no incentives to participate other than the provision of free health care at the study clinic, clinic travel expenses and an ITN. Written informed consent was obtained in the appropriate language from guardians of children and from adult participants and assent was requested from children 8–17 years of age as per Uganda National Council for Science and Technology guidelines. Written informed consent and assent forms were available in all local languages prevalent in Tororo district (English, Japadhola, Kiswahili, Lusamya and Luganda). If the adult, child, or parent/guardian was unable to read, an independent witness was identified. Approval from local leaders was obtained before beginning activities. Ethics approval was provided by the Uganda National Council for Science and Technology; Makerere University School of Medicine Research and Ethics Committee; University of California, San Francisco Committee for Human Research; Department of Biosciences Ethics Committee, Durham University; and the London School of Hygiene & Tropical Medicine Ethics Committee.

## Results

### Characteristics of participants and their households

A total of 531 participants (378 children and 153 adults) residing in 80 households were enrolled in the cohort and followed for a total of 958.1 person years (Fig 1). The mean age of participants was 16.8 years (95% confidence interval (CI): 15.4–18.2 years) and 253 (47.7%) were male (Table 1). Reported ITN use the night before, as ascertained at clinic visits, was 96%

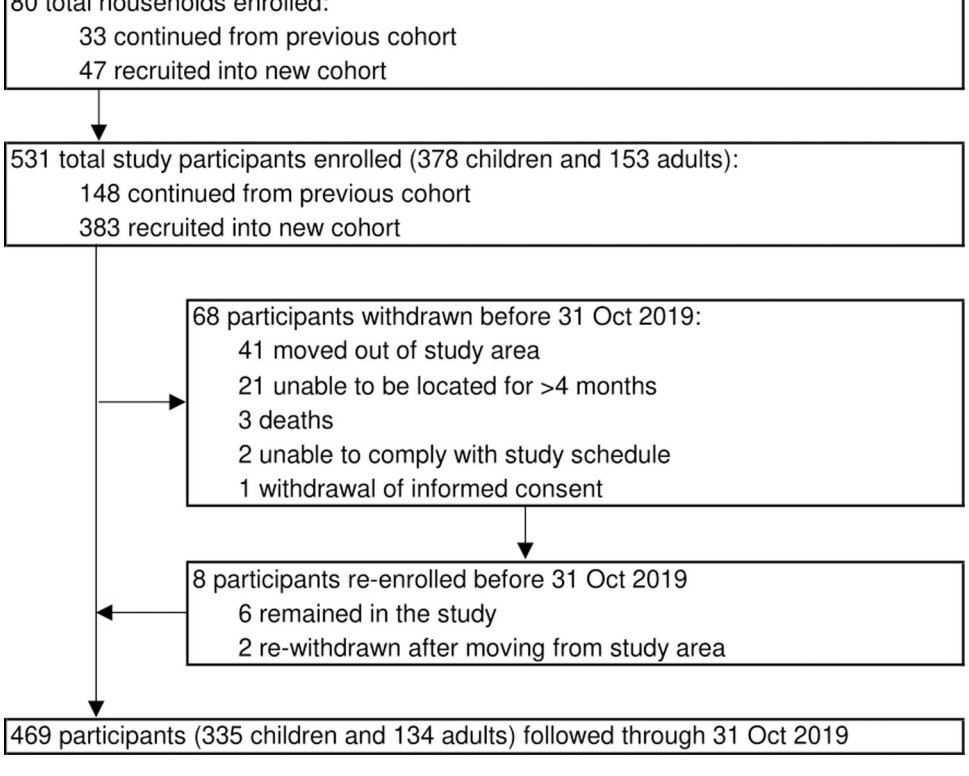

**Fig 1. Study flow.**

**Table 1. Characteristics of study participants and households in Nagongera, Uganda.**

| Characteristic | | All households | House type [a] | | |
| --- | --- | --- | --- | --- | --- |
| | | | Traditional | Modern | p value |
| *Individual study participant data (n = 531)* | | | | | |
| Male (%) | | 47.7 (253) | 48.0 (140) | 47.3 (113) | 0.88 |
| Mean age during follow up (%) | <5 years | 27.1 (144) | 27.4 (80) | 26.8 (64) | 0.85 |
| | 5–15 years | 41.8 (222) | 40.8 (119) | 43.1 (103) | |
| | >15 years | 31.1 (165) | 31.9 (93) | 30.1 (72) | |
| *Individual household level data (n = 80)* | | | | | |
| Wealth category (%) | Poorest | 33.8 (27) | 44.4 (20) | 20.0 (7) | 0.06 |
| | Middle | 33.8 (27) | 31.1 (14) | 37.1 (13) | |
| | Least poor | 32.5 (26) | 24.4 (11) | 42.9 (15) | |
| IRS in the past 12 months (%) | No | 10.0 (8) | 11.1 (5) | 8.6 (3) | 0.71 |
| | Yes | 90.0 (72) | 88.9 (40) | 91.4 (32) | |
| Main roof material (%) | Thatched | 28.8 (23) | 51.1 (23) | 0.0 (0) | <0.001 |
| | Metal | 71.3 (57) | 48.9 (22) | 100.0 (35) | |
| Main floor material (%) | Earth, sand or dung | 91.3 (73) | 100.0 (45) | 80.0 (28) | 0.002 |
| | Cement or concrete | 8.8 (7) | 0.0 (0) | 20.0 (7) | |
| Type of eaves (%) | Open | 31.3 (25) | 55.6 (25) | 0.0 (0) | <0.001 |
| | Closed | 68.8 (55) | 44.4 (20) | 100.0 (35) | |
| Windows per room (%) | 0 window | 43.8 (35) | 55.6 (25) | 28.6 (10) | 0.02 |
| | 0<1 window | 30.0 (24) | 28.9 (13) | 31.4 (11) | |
| | 1 window | 26.3 (21) | 15.6 (7) | 40.0 (14) | |
| Airbricks present (%) | Yes | 42.5 (34) | 71.1 (32) | 5.7 (2) | <0.001 |
| | No | 57.5 (46) | 28.9 (13) | 94.3 (33) | |
| People per bedroom (%) | ≥3 people | 70.0 (56) | 80.0 (36) | 57.1 (20) | 0.03 |
| | 0–2 people | 30.0 (24) | 20.0 (9) | 42.9 (15) | |
| Sanitation facility [b] (%) | Unimproved | 91.3 (73) | 91.1 (41) | 91.4 (32) | 0.96 |
| | Improved | 8.8 (7) | 8.9 (4) | 8.6 (3) | |
| Drinking water source [c] (%) | Unimproved | 33.8 (27) | 31.1 (14) | 37.1 (13) | 0.57 |
| | Improved | 66.3 (53) | 68.9 (31) | 62.9 (22) | |

IRS: Indoor residual spraying.

[a] Modern houses: Closed eaves, brick (not mud walls), metal (not thatched) roof; traditional houses: All other houses.

[b] Sanitation facility was defined as improved or unimproved using WHO Joint Monitoring Programme (WHO-JMP) criteria which consider whether or not a sanitation facility adequately separates human excreta from human contact (improved toilets include latrines with washable slabs) [30].

[c] Drinking water source was defined as improved or unimproved using WHO Joint Monitoring Programme (WHO-JMP) criteria which consider whether or not a drinking water source has adequate protection from outside contamination (improved sources include piped water, protected wells and rainwater) [30].

(11,619 of 12,165 routine visits), but only 70% (56 of 80) households had at least one ITN per two people at baseline. Overall, 56% (45) of houses were classified as traditional (Fig 2). The median number of rooms used for sleeping was two rooms (range 1–3) and median household size was six people (range 3–8), with 70% (56) households having three or more residents per room. Sixty-six percent (53) of houses had windows, 69% (55) had closed eaves and the most common roof material was metal sheets, recorded in 71% (57) of houses. In the wealth index, the first principal component explained 21% of overall variability in the asset variables. The weight assigned to each variable was: bed (0.49), table (0.48), acres of agricultural land (0.39), bicycle (0.36), radio (0.32), mobile telephone (0.30), type of toilet facility (0.22), type of drinking water (0.04), mode of transport to the health facility (0.01).

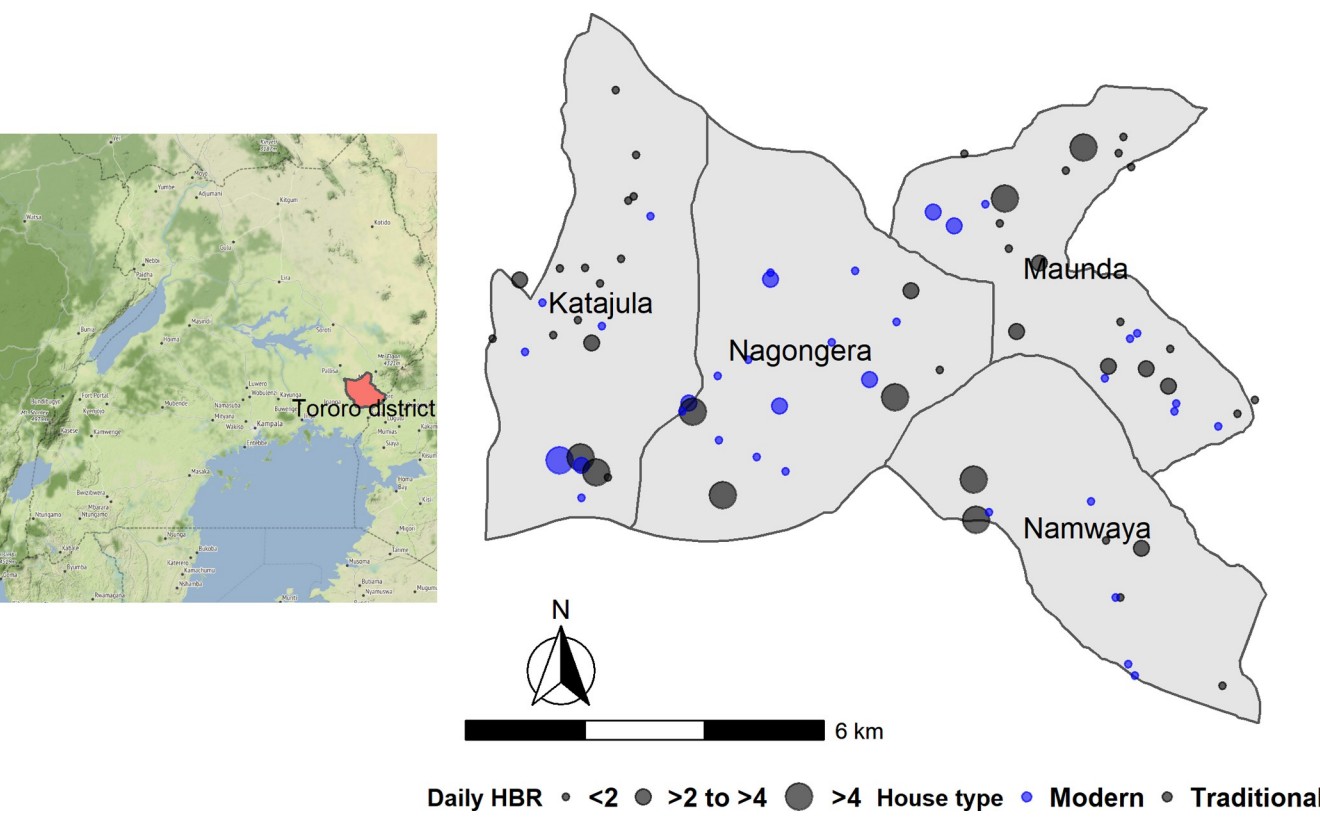

**Fig 2. Map of the study area showing distribution of house type and daily human biting rate (base map source: http://maps.stamen.com/).**

## Association between house type and entomological and epidemiological outcomes

**Human biting rate (HBR).** A total of 15,780 adult female *Anopheles* mosquitoes were collected over 24 months and 7,631 collection nights, giving an overall HBR of 2.07 *Anopheles* per room per night. Of these, 98% (15,462 of 15,780) were *An. gambiae* s.l. giving an HBR of 2.03 *An. gambiae* per room per night (of which 99% [1310 of 1325] were *An. arabiensis*); 0.2% (38 of 15,780) were *An. funestus* giving an HBR of 0.005 *An. funestus* per room per night and 1.8% (280 of 15,780) were other *Anopheles* species giving a HBR of 0.04 other *Anopheles* per room per night. A total of 72 female *Culex* spp. were collected. Nine of 15,778 *Anopheles* tested for sporozoites were positive, giving an overall sporozoite rate of 0.06% and an annual EIR of 0.43 infectious bites per room. Peak biting rates occurred a few weeks after the peak rains (Fig 3).

HBR was consistently lower in rooms with modern house features than traditional features (Tables 2 and S2). Adjusting for household wealth, IRS in the past 12 months and ITN use, HBR was 53% lower in modern rooms compared to traditional rooms (adjusted incidence rate ratio (aIRR) 0.47, 95% CI 0.32–0.67, p<0.001); 60% lower in rooms with metal roofs compared with thatch (aIRR 0.40, 95% CI 0.24–0.68, p = 0.001) and 60% lower in rooms with closed eaves compared to open eaves (aIRR 0.40, 95% CI 0.25–0.65, p<0.001). HBR was 30% lower in rooms with windows, compared to rooms with no windows, although this difference was of borderline significance (aIRR 0.70, 95% CI 0.49–1.00, p = 0.05), and 49% lower in rooms with airbricks, compared to rooms with no airbricks (aIRR 0.51, 95% CI 0.35–0.73, p<0.001). There was no association between HBR and household wealth, IRS, LLIN use, crowding or the average time residents reported going to bed.

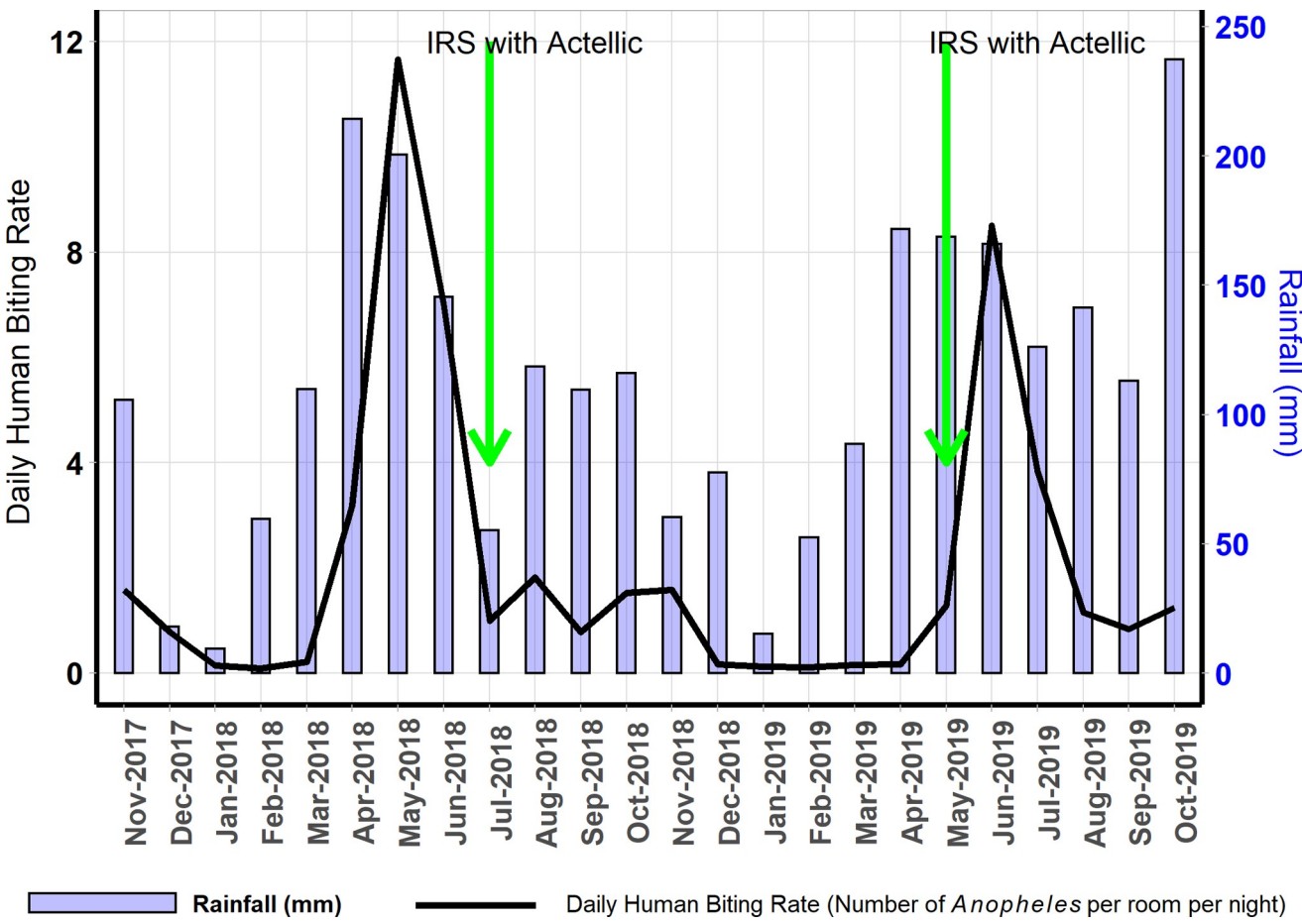

**Fig 3. Mean daily human biting rate by month, in relation to two indoor residual spraying campaigns with Actellic and mean monthly rainfall estimated for Nagongera sub-county using the National Aeronautics and Space Administration Tropical Rainfall Measuring Mission [29].** Peak biting rates correspond with higher rainfall.

### Microscopic and sub-microscopic parasitaemia and malaria incidence

Of 13,277 blood smears taken, 277 (2.1%) had malaria parasites detected by microscopy. Of 12,728 blood samples assessed for parasitaemia by qPCR, 1,343 (10.6%) were positive, giving an overall prevalence of microscopic or sub-microscopic malaria infection of 10.2% (1,347 of 13,277 blood samples). All participants contributed at least one blood sample. Unadjusted results are presented in S3 Table. In the adjusted analysis, the odds of any malaria infection were 72% higher among males than females (adjusted odds ratio (aOR) 1.72, 95% CI 1.02–2.88; p = 0.04) and three- to fourfold higher among older participants compared to children aged <5 years (5–15 years: aOR 3.96, 95% CI 2.27–6.90, p<0.001; >15 years: aOR 3.15, 95% CI 1.70–5.84, p<0.001) (Table 3). Malaria infection was not associated with ITN use reported at the clinic visit, household wealth or any housing characteristics. Over 958 total person years, 38 episodes of malaria were diagnosed during the study period, giving an overall incidence of 0.04 episodes per person-year at risk. There were no cases of severe malaria or malaria deaths, although two episodes of malaria with danger signs were documented in children younger than 5 years (vomiting and lethargy in a one year-old child and lethargy and inability to sit upright in a three year-old child). No association was found between malaria incidence and household-level factors in either the unadjusted or adjusted analysis (Tables 3 and S3).

**Table 2. Association between room characteristics and malaria vector density in Nagongera, Uganda (adjusted results).**

| Characteristic | | HBR [a] (total collection nights) | IRR [b] (95% CI) | p value |
|---|---|---|---|---|
| *Household-level characteristics* | | | | |
| Wealth category | Poorest | 1.91 (2342) | 1 | 0.68 |
| | Middle | 2.09 (2445) | 1.28 (0.69, 2.37) | |
| | Least poor | 2.18 (2844) | 1.17 (0.73, 1.89) | |
| IRS in the past 12 months | No | 2.46 (635) | 1 | 0.49 |
| | Yes | 2.03 (6996) | 0.80 (0.43, 1.49) | |
| People per bedroom | ≥3 people | 2.11 (5013) | 1 | 0.69 |
| | 0–2 people | 2.00 (2618) | 0.91 (0.55, 1.49) | |
| *Room-level characteristics* | | | | |
| ITN use in the room | <50% of nights | 2.38 (3855) | 1 | 0.07 |
| | ≥50% of nights | 1.75 (3776) | 0.69 (0.45, 1.04) | |
| House type [c] | Traditional | 2.98 (3251) | 1 | <0.001 |
| | Modern | 1.39 (4380) | 0.47 (0.32, 0.67) | |
| Main roof material | Thatched | 3.75 (1398) | 1 | 0.001 |
| | Metal | 1.69 (6233) | 0.40 (0.24, 0.68) | |
| Type of eaves | Open | 3.68 (1602) | 1 | <0.001 |
| | Closed | 1.64 (6029) | 0.40 (0.25, 0.65) | |
| Windows present | No | 2.41 (4083) | 1 | 0.05 |
| | Yes | 1.67 (3548) | 0.70 (0.49, 1.00) | |
| Airbricks present | No | 2.89 (3351) | 1 | <0.001 |
| | Yes | 1.43 (4280) | 0.51 (0.35, 0.73) | |
| Average time to bed | 2100 or later | 2.16 (2626) | 1 | 0.94 |
| | Before 2100 | 2.02 (5005) | 0.99 (0.76, 1.28) | |

CI: Confidence interval, HBR: Human biting rate, IRR: Incidence rate ratio, IRS: Indoor residual spraying, ITN: Long-lasting insecticide-treated net.

[a] Total adult female *Anopheles* caught per room per night.

[b] Adjusted for household wealth category, IRS in the past 12 months, ITN use the night of the entomology survey.

[c] Modern houses: Closed eaves, brick (not mud walls), metal (not thatched) roof; traditional houses: All other houses.

## Acute respiratory infection (ARI)

A total of 2,553 episodes of ARI were diagnosed over 958 person years, giving an overall incidence of 2.7 episodes per person-year. Adjusting for age and household wealth, incidence of ARI was 11% lower in males compared to females (aIRR 0.89, 95% CI 0.82–0.98, p = 0.01) and adjusting for gender and household wealth, ARI incidence was 49% to 53% lower among older participants compared to children aged <5 years (5–15 years: aIRR 0.51, 95% CI 0.46–0.57, p<0.001; >15 years: aIRR 0.47, 95% CI 0.41–0.54, p<0.001) (Table 4). There was no association between ARI and any household characteristics.

## Gastrointestinal illness

A total of 387 episodes of gastrointestinal illness were diagnosed over 958 person years, giving an overall incidence of 0.4 episodes per person-year. Adjusting for gender, household wealth, sanitation facility and drinking water source, incidence of gastrointestinal illness was 77% lower among participants aged 5–15 years, compared to children aged <5 years (aIRR 0.23, 95% CI 0.16–0.33, p<0.001) but there was no difference among participants aged >15 years. Gastrointestinal illness incidence was 24% lower in modern houses compared to traditional houses (aIRR 0.76, 95% CI 0.59–0.98, p = 0.04), 31% lower among households with one

**Table 3. Association between house type and malaria in Nagongera, Uganda (adjusted results).**

| Characteristic | | PR (total blood slides) | OR [a] (95% CI) | p value | Malaria incidence (total person years) | IRR [a] (95% CI) | p value |
|---|---|---|---|---|---|---|---|
| *Individual-level characteristics* | | | | | | | |
| Gender | Female | 7.8 (6901) | 1 | 0.04 | 0.04 (498.9) | 1 | 0.31 |
| | Male | 12.8 (6326) | 1.72 (1.02, 2.88) | | 0.03 (459.2) | 0.73 (0.39, 1.35) | |
| Age [b] | <5 years | 3.9 (3663) | 1 | <0.001 | 0.04 (249.0) | 1 | 0.06 |
| | 5–15 years | 13.9 (5450) | 3.96 (2.27, 6.90) | | 0.05 (402.7) | 1.18 (0.58, 2.39) | |
| | >15 years | 10.8 (4114) | 3.15 (1.70, 5.84) | | 0.02 (306.4) | 0.44 (0.17, 1.17) | |
| ITN use the night before clinic visit | No | 9.2 (563) | 1 | 0.51 | - | - | - |
| | Yes | 9.9 (12133) | 1.12 (0.79, 1.60) | | - | - | |
| ITN use the night before clinic visit | <90% of visits | - | - | - | 0.03 (104.8) | 1 | 0.51 |
| | ≥90% of visits | - | - | | 0.04 (853.3) | 1.46 (0.47, 4.51) | |
| *Household-level characteristics* | | | | | | | |
| Wealth category | Poorest | 11.1 (4506) | 1 | 0.81 | 0.05 (326.2) | 1 | 0.19 |
| | Middle | 9.1 (4266) | 0.80 (0.40, 1.59) | | 0.05 (308.7) | 0.85 (0.41, 1.74) | |
| | Least poor | 10.2 (4455) | 0.92 (0.44, 1.95) | | 0.02 (323.2) | 0.43 (0.17, 1.10) | |
| IRS in past 12 months | No | 7.1 (1418) | 1 | 0.26 | 0.04 (101.4) | 1 | 0.97 |
| | Yes | 10.6 (11809) | 1.67 (0.68, 4.10) | | 0.04 (856.7) | 1.02 (0.40, 2.57) | |
| House type [c] | Traditional | 9.8 (7169) | 1 | 0.77 | 0.04 (517.8) | 1 | 0.86 |
| | Modern | 10.7 (6058) | 1.11 (0.56, 2.17) | | 0.04 (440.3) | 0.94 (0.51, 1.74) | |
| Main roof material | Thatched | 8.0 (3516) | 1 | 0.31 | 0.04 (254.6) | 1 | 0.82 |
| | Metal | 11.0 (9711) | 1.43 (0.72, 2.83) | | 0.04 (703.5) | 1.09 (0.52, 2.31) | |
| Type of eaves | Open | 8.9 (3885) | 1 | 0.51 | 0.04 (281.0) | 1 | 0.88 |
| | Closed | 10.7 (9342) | 1.24 (0.66, 2.31) | | 0.04 (677.1) | 1.05 (0.51, 2.16) | |
| Windows per room | 0 window | 7.3 (5574) | 1 | 0.15 | 0.04 (406.5) | 1 | 0.30 |
| | 0<1 window | 12.8 (4159) | 1.83 (0.96, 3.46) | | 0.03 (298.8) | 0.71 (0.30, 1.71) | |
| | 1 window | 11.6 (3494) | 1.74 (0.79, 3.80) | | 0.05 (252.8) | 1.49 (0.69, 3.23) | |
| Airbricks present | No | 7.9 (5300) | 1 | 0.26 | 0.03 (382.8) | 1 | 0.18 |
| | Yes | 11.7 (7927) | 1.51 (0.73, 3.12) | | 0.05 (575.3) | 1.62 (0.80, 3.29) | |
| People per bedroom | ≥3 people | 11.1 (9560) | 1 | 0.17 | 0.04 (693.3) | 1 | 0.92 |
| | 0–2 people | 7.9 (3667) | 0.61 (0.31, 1.22) | | 0.03 (264.8) | 1.04 (0.50, 2.15) | |

CI: Confidence interval, IRS: Indoor residual spraying, ITN: Insecticide treated net, OR: Odds ratio, PR: Parasite rate.

[a] Adjusted for age, gender, ITN use, IRS in the past 12 months, household wealth category.

[b] Malaria prevalence: Age at the time of the routine clinic visit; malaria incidence: Mean age during follow up.

[c] Modern houses: Closed eaves, brick (not mud walls), metal (not thatched) roof; traditional houses: All other houses.

**Table 4. Association between house type and incidence of acute respiratory infection and gastrointestinal illness in Nagongera, Uganda (adjusted results).**

| Characteristic | | ARI | | | Gastrointestinal illness | | |
|---|---|---|---|---|---|---|---|
| | | Incidence (total person years) | IRR [b] (95% CI) | p value | Incidence (total person years) | IRR [c] (95% CI) | p value |
| *Individual-level characteristics* | | | | | | | |
| Gender | Female | 2.8 (498.9) | 1 | 0.01 | 0.5 (498.9) | 1 | 0.11 |
| | Male | 2.5 (459.2) | 0.89 (0.82, 0.98) | | 0.3 (459.2) | 0.80 (0.60, 1.06) | |
| Mean age during follow up | <5 years | 4.3 (249.0) | 1 | <0.001 | 0.5 (249.0) | 1 | <0.001 |
| | 5–15 years | 2.2 (402.7) | 0.51 (0.46, 0.57) | | 0.1 (402.7) | 0.23 (0.16, 0.33) | |
| | >15 years | 2.0 (306.4) | 0.47 (0.41, 0.54) | | 0.7 (306.4) | 1.20 (0.90, 1.61) | |
| *Household-level characteristics* | | | | | | | |
| Wealth category | Poorest | 2.6 (326.2) | 1 | 0.09 | 0.4 (326.2) | 1 | 0.79 |
| | Middle | 3.0 (308.7) | 1.16 (1.00, 1.35) | | 0.4 (308.7) | 0.96 (0.68, 1.36) | |
| | Least poor | 2.5 (323.2) | 0.97 (0.79, 1.18) | | 0.4 (323.2) | 0.89 (0.62, 1.27) | |
| House type [d] | Traditional | 2.8 (517.8) | 1 | 0.30 | 0.5 (517.8) | 1 | 0.04 |
| | Modern | 2.5 (440.3) | 0.92 (0.77, 1.08) | | 0.3 (440.3) | 0.76 (0.59, 0.98) | |
| Windows per room | 0 window | 2.6 (406.5) | 1 | 0.72 | 0.4 (406.5) | 1 | <0.001 |
| | 0<1 window | 2.8 (298.8) | 1.07 (0.91, 1.24) | | 0.5 (298.8) | 1.19 (0.89, 1.60) | |
| | 1 window | 2.6 (252.8) | 1.02 (0.84, 1.23) | | 0.3 (252.8) | 0.69 (0.51, 0.93) | |
| Airbricks present | No | 2.6 (382.8) | 1 | 0.52 | 0.5 (382.8) | 1 | 0.24 |
| | Yes | 2.7 (575.3) | 1.05 (0.91, 1.21) | | 0.4 (575.3) | 0.85 (0.65, 1.12) | |
| Main floor material | Earth, sand or dung | 2.7 (867.0) | 1 | 0.55 | 0.4 (867.0) | 1 | 0.28 |
| | Cement or concrete | 2.6 (91.1) | 0.91 (0.68, 1.23) | | 0.4 (91.1) | 0.80 (0.54, 1.19) | |
| People per bedroom | ≥3 people | 2.6 (693.3) | 1 | 0.34 | 0.4 (693.3) | 1 | 0.74 |
| | 0–2 people | 2.8 (264.8) | 1.09 (0.91, 1.30) | | 0.4 (264.8) | 1.06 (0.77, 1.44) | |
| Sanitation facility [e] | Unimproved | 2.6 (878.3) | 1 | 0.17 | 0.4 (878.3) | 1 | 0.02 |
| | Improved | 3.5 (79.8) | 1.32 (0.89, 1.98) | | 0.7 (79.8) | 1.66 (1.09, 2.53) | |
| Drinking water source [f] | Unimproved | 2.8 (330.3) | 1 | 0.11 | 0.4 (330.3) | 1 | 0.10 |
| | Improved | 2.6 (627.8) | 0.90 (0.79, 1.02) | | 0.4 (627.8) | 0.79 (0.60, 1.05) | |

ARI: Acute respiratory infection, CI: Confidence interval, ITN: Insecticide treated net, IRR: Incidence rate ratio, IRS: Indoor residual spraying.

[a] Adjusted for age, gender, ITN use, IRS in the past 12 months, household wealth category.

[b] Adjusted for age, gender, household wealth category.

[c] Adjusted for age, gender, household wealth category, sanitation facility, drinking water source.

[d] Modern houses: Closed eaves, brick (not mud walls), metal (not thatched) roof; traditional houses: All other houses.

[e] Sanitation facility was defined as improved or unimproved using WHO Joint Monitoring Programme (WHO-JMP) criteria which consider whether or not a sanitation facility adequately separates human excreta from human contact (improved toilets include latrines with washable slabs) [30].

[f] Drinking water source was defined as improved or unimproved using WHO Joint Monitoring Programme (WHO-JMP) criteria which consider whether or not a drinking water source has adequate protection from outside contamination (improved sources include piped water, protected wells and rainwater) [30].

window per room, compared to no windows (aIRR 0.69, 95% CI 0.51–0.93, p = 0.01) and 66% higher among households with an improved, compared to unimproved latrine (aIRR 1.66, 95% CI 1.09–2.53, p = 0.02) (Table 4). There was no association between gastrointestinal illness and household wealth, crowding and type of drinking water source in either the unadjusted or adjusted analysis (Tables 4 and S4).

## Discussion

We identified and quantified a relationship between house design and malaria, ARI and gastrointestinal illness in Nagongera, Uganda, a rural area where sustained vector control has reduced malaria transmission from high to low endemicity. Modern houses (defined as houses with three features: brick walls, metal roofs and closed eaves) were associated with a 53% lower HBR and a 24% lower incidence of gastrointestinal illness compared to traditional houses (all other homes), but no association with malaria prevalence and incidence nor ARI incidence, after controlling for factors including age, gender, household wealth, ITN use, IRS and type of sanitation and drinking water source. Our findings suggest that house improvements may contribute to the maintenance of low malaria endemicity in rural communities where transmission has declined following intensive vector control, with concurrent benefits for the prevention of gastrointestinal illness among children and adults.

The present study adds to a series of investigations of the relationship between housing and malaria in Nagongera between 2011 and 2017 [14–16]. Before the scale-up of IRS, modern housing was associated with a 52% lower HBR during 2011–2013 [15] and a 48% lower HBR during 2011–2015 [16] compared to traditional housing; following IRS scale-up this strength of association increased to a 73% lower HBR during 2015–2017 [16]. The 53% lower HBR associated with modern housing in the present study is consistent with the previous findings, confirming the potential for house design to reduce human exposure to infectious bites by mosquitoes after malaria transmission has declined to very low levels. These findings support those from other settings, including a meta-analysis of nine randomized studies that found a 65% average reduction in indoor densities of both *Aedes* and *Anopheles* associated with house improvements in Africa and Latin America [31]. We found no association between house design and malaria infection, in contrast to previous observations in the cohort study of 41% lower odds of infection during 2011–2013 and 57% lower odds of infection during 2015–17 in modern houses, compared to traditional houses [16]. It is possible that this reflects a slowing in the rate of decline of *Pf*PR following interruption of transmission as baseline prevalence reduces [32]. Additionally, ITNs and IRS may have shifted more malaria transmission outdoors in Nagongera, thereby blunting any benefits of housing improvements by reducing populations of the highly endophilic and anthrophilic *An. gambiae s.s.* in favour of the more exophilic and zoophilic *An. arabiensis* [33]. Finally, particularly when mosquito densities are perceived to be lower, people may keep doors open until midnight allowing vectors to enter houses [34]. Our observation of no association between house design and malaria incidence is consistent with earlier findings [15,16].

Modern housing, with metal roofs, brick walls and closed eaves, may reduce exposure to malaria vectors in several ways. First, house entry by *An. gambiae* is likely to decline when eaves are closed and when homes have fewer access points and resting sites for mosquitoes [35]; in the present study we observed that HBR was 60% lower in bedrooms with closed rather than open eaves. Second, metal-roofed houses are typically hotter than thatch-roofed houses in the daytime, which may reduce survival of both the vector [36] and parasite [37]. Third, $CO_2$ mediates the relationship between house design and indoor vector densities. Indoor $CO_2$ concentration may be lower in houses that have thatch roofs rather than metal, are raised

rather than at ground level, or that have a relatively high window surface area; this reduces the attractiveness of the building to mosquitoes [38,39]. In the present study, HBR was 30% lower in bedrooms with windows compared to no windows (although this was of borderline significance) and 49% lower in bedrooms with airbricks compared to no airbricks, consistent with a build-up of $CO_2$ due to low ventilation and a consequent influx of mosquitoes into these rooms from elsewhere. Windows and airbricks, however, were more common in modern than traditional homes, making it difficult to disentangle the direct effects of windows and airbricks from indirect effects of other housing features. Measurement of indoor temperature, airflow and $CO_2$ within Ugandan houses would improve understanding of these mechanisms.

Modern housing was associated with a 24% reduction in incidence of gastrointestinal illness, compared to traditional housing. Our definition of gastrointestinal illness was broad, encompassing acute diarrhoea, gastroenteritis, helminth infection and other diagnoses (S1 Table). Diarrhoea prevention is largely based on the reduction of faecal-oral transmission of pathogens via an improved water supply (ideally a piped connection into the home), improved sanitation facility (ideally a flush toilet to a confined system) and handwashing with soap [40]. Water and sanitation are of higher quality in modern houses than traditional houses in many settings [9] but not in our study area, suggesting that housing and gastrointestinal illness were linked by another mechanism. Greenbottle flies from latrines have been identified as a putative vector of diarrheal disease; better constructed houses may therefore provide protection [13]. Solid floors have a lower risk of helminth infection [41] and we found that floor type was strongly associated with house design (every traditional home had an earth, sand or dung floor, rather than concrete or cement), although there was no association between floor type and gastrointestinal illness. Further investigations of the mechanisms linking house design and gastrointestinal illness are needed. Surprisingly, we found a 66% increase in gastrointestinal illness among participants with an improved, compared to an unimproved toilet. We classified improved toilets as those considered to adequately separate human excreta from human contact, including covered pit latrines with a slab, while unimproved facilities included covered pit latrines with no slab, uncovered pit latrines and no facility. Transmission of diarrhoea by greenbottle flies emerging from latrines is possible [13] but it is unclear why covered latrines were associated with an increase in transmission. Latrines may be foci of disease transmission where multiple households share the facility [42], but sharing of latrines was not measured in our study population. Since the "improved" sanitation category contained only seven households, our findings require replication on a larger scale.

We found no association between house design and ARI, which are expected to be linked via indoor air quality and ventilation; indoor air pollution from solid fuels is estimated to cause more than half of pneumonia deaths globally in young children [43]. There may also be an effect of crowding on ARI but the evidence is mixed [44]. Our observation of no association between house design and ARI is consistent with a recent meta-analysis of 155,609 children in 55 national surveys, which similarly found no link [7]. This may reflect no underlying association, or a complex pattern of factors such as inhalation of cooking smoke both indoors and outdoors that offsets any effect of house design and poor ventilation in most houses regardless of design. ARI remains the leading cause of death in children aged 1–5 years globally and, in the context of the COVID-19 pandemic, it is important to consider how house design affects this health outcome.

While house design was the primary exposure in this study, no association was found between ITN use nor IRS and any malaria outcomes. This finding contrasts with earlier observations that suggest good protective efficacy of both ITNs and IRS in the same study population. Previously, IRS was associated with a 98% reduction in age-adjusted malaria incidence after five to six rounds, compared to before any IRS [22] and during the first 18

months of the present study, individuals sleeping in rooms with two or fewer mosquitoes caught per night had three times the odds of not using ITNs compared to individuals sleeping in rooms with three or more mosquitoes caught per night [45]. A threefold increase in the odds of a malaria episode was also found among participants who did not report always using their ITN in the prior 1–5 weeks, compared to participants who always used their ITN [45]. The earlier analysis of ITNs [45] was stratified by time period and found no association in the final six months when both mosquito densities and ITN use in all age groups markedly increased. We used a different ITN metric (ITN use the night before the diagnosis as reported at the study clinic, rather than ITN use recorded the night before the entomology visit) which may have overestimated true ITN use. The methods of analysis also differed (the earlier analysis used a case-control design) as did the variables adjusted for (the present study controlled for household wealth, age and gender but these were not included in the earlier model). Taken together, the findings of both studies highlight the importance of ITN use, IRS and house design as mosquito densities fall, and the need to better understand interactions between the three.

With urbanisation, population expansion and economic growth, housing is widely changing across SSA, offering major potential for improvements in health [9]. This trend is evident in Nagongera where, between 2013 and 2016, 13% of surveyed households replaced thatched roofs with metal roofs, 31% built or modified homes to replace mud with brick walls and 19% closed the eaves of their homes [16]. Changes in housing may have contributed to the reductions in malaria observed in the study area since 2013, either directly by reducing exposure to vectors or indirectly by enhancing the effects of IRS (since residual insecticides often have greater effectiveness on impervious surfaces, compared to porous mud and dung surfaces [46]). Modern housing may also be helping to maintain low endemicity in a setting where repeated rounds of IRS have significantly reduced transmission [33], which would point to housing improvements as a potentially sustainable intervention in elimination settings. Effectiveness, however, needs to be better understood where the outdoor-biting *An. arabiensis* dominates transmission. Being community-led and delivered, as well as a human right, housing improvements offer the advantage of a 'development' intervention that benefits a range of health and social needs [3]. Our study suggests that the health co-benefits of housing improvements include prevention of gastrointestinal illness, but future research is needed to determine causality and protective mechanisms.

The study has several limitations. First, the observational design makes it difficult to determine causality. Indeed, a high disease burden may reduce income, thereby providing a reverse pathway to poorer housing. Similarly, residual confounding by household wealth of the relationship between housing and disease outcomes is possible. Second, the ITN use metric analysed in relation to clinical malaria outcomes may have overestimated ITN use. Third, it would have been useful to assess whether season (wet *versus* dry) was an effect modifier of the association between house design and the epidemiological outcomes, but this was precluded by low incidence of these outcomes. Fourth, the findings may not be generalisable outside the cohort population, which was subject to intense observation. Nonetheless our observations are consistent with an increasing weight of evidence that house design is an important determinant of health in SSA [7,10].

In conclusion, community-led changes to housing in Uganda may contribute to the maintenance of low malaria endemicity in rural areas where transmission has declined following intensive vector control, with concurrent benefits for reducing gastrointestinal illness among children and adults. Further studies are needed to investigate causality, biological modes of action and impact in different settings.

## Supporting information

**S1 Table. Diagnoses included in the definitions of acute respiratory infection and gastrointestinal illness.**
(DOCX)

**S2 Table. Association between room characteristics and malaria vector density in Nagongera, Uganda (unadjusted results).**
(DOCX)

**S3 Table. Association between house type and malaria in Nagongera, Uganda (unadjusted results).**
(DOCX)

**S4 Table. Association between house type and incidence of acute respiratory infection and diarrhoeal disease in Nagongera, Uganda (unadjusted results).**
(DOCX)

**S1 Text. Checklist of items that should be included in reports of cohort studies.**
(DOCX)

**S2 Text. Questionnaire for inclusivity in global research.**
(DOCX)

## Acknowledgments

We thank all study participants. We also thank specific household members who volunteered to switch on and off the light traps.

## Author Contributions

**Conceptualization:** Joaniter I. Nankabirwa, Moses R. Kamya, Sarah G. Staedke, Chris Drakeley, Teun Bousema, Steve W. Lindsay, Grant Dorsey.

**Data curation:** Alex K. Musiime, Paul J. Krezanoski, Geoffrey Otto, Patrick Kyagamba, Jackson Asiimwe, John Rek, Lucy S. Tusting.

**Formal analysis:** Alex K. Musiime, Paul J. Krezanoski, Grant Dorsey, Lucy S. Tusting.

**Funding acquisition:** Joaniter I. Nankabirwa, Moses R. Kamya, Sarah G. Staedke, Chris Drakeley, Teun Bousema, Steve W. Lindsay, Grant Dorsey.

**Investigation:** Alex K. Musiime, Joaniter I. Nankabirwa, Emmanuel Arinaitwe, Moses R. Kamya, Sarah G. Staedke, Chris Drakeley, Teun Bousema, Steve W. Lindsay, Grant Dorsey.

**Methodology:** Anne M. Akol.

**Project administration:** Paul J. Krezanoski, John Rek, Joaniter I. Nankabirwa, Emmanuel Arinaitwe, Moses R. Kamya, Sarah G. Staedke, Grant Dorsey.

**Supervision:** Alex K. Musiime, Maxwell Kilama, John Rek, Emmanuel Arinaitwe, Moses R. Kamya.

**Visualization:** Alex K. Musiime.

**Writing – original draft:** Alex K. Musiime, Paul J. Krezanoski, David L. Smith, Melissa D. Conrad, Lucy S. Tusting.

**Writing – review & editing:** Alex K. Musiime, Paul J. Krezanoski, David L. Smith, Maxwell Kilama, Melissa D. Conrad, Geoffrey Otto, Patrick Kyagamba, Jackson Asiimwe, John Rek, Joaniter I. Nankabirwa, Emmanuel Arinaitwe, Anne M. Akol, Moses R. Kamya, Sarah G. Staedke, Chris Drakeley, Teun Bousema, Steve W. Lindsay, Grant Dorsey, Lucy S. Tusting.

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
