## [Decision Letter · Decision Letter 0]

8 Oct 2021

PGPH-D-21-00703

House design and risk of malaria, acute respiratory infection and gastrointestinal illness in Uganda: a cohort study

Dear Mr. MUSIIME,

Thank you for submitting your manuscript to PLOS Global Public Health. After careful consideration, we feel that it has merit but does not fully meet PLOS Global Public Health’s publication criteria as it currently stands. Therefore, we invite you to submit a revised version of the manuscript that addresses the points raised during the review process.

On a general note, the subject of study in the manuscript has been evaluated as very important for global health interest. However, they have made recommendations for corrections, particularly, for clarity of intentions and methodology as pointed out by reviewer_1 and _3

We look forward to receiving your revised manuscript.

Kind regards,

Nnodimele Onuigbo Atulomah, PhD

Academic Editor

Journal Requirements:.

2. Please provide separate figure files in .tif or .eps format only, and remove any figures embedded in your manuscript file.  If you are using LaTeX, you do not need to remove embedded figures.

3. As the link do not return a result, please amend your Data Availability Statement and indicate where the data may be found.

4. Please provide us with a direct link to the base layer of the map used in Figure 2 and ensure this location is also included in the figure legend. 

Please note that, because all PLOS articles are published under a CC BY license (creativecommons.org/licenses/by/4.0/), we cannot publish proprietary maps such as Google Maps, Mapquest or other copyrighted maps. If your map was obtained from a copyrighted source please amend the figure so that the base map used is from an openly available source.

Please note that only the following CC BY licences are compatible with PLOS licence: CC BY 4.0, CC BY 2.0  and CC BY 3.0, meanwhile such licences as CC BY-ND 3.0 and others are not compatible due to additional restrictions. If you are unsure whether you can use a map or not, please do reach out and we will be able to help you. 

The following websites are good examples of where you can source open access or public domain maps:

5. Please ensure that the funders and grant numbers match between the Financial Disclosure field and the Funding Information tab in your submission form. Note that the funders must be provided in the same order in both places as well.

Additional Editor Comments (if provided):

Your manuscript "House design and risk of malaria, acute respiratory infection and gastrointestinal illness in Uganda: a cohort study" (PGPH-D-21-00703) has been assessed by our reviewers. They have raised a few points which we believe would improve the manuscript and may allow a revised version to be published in PLoS Global Public Health.

Reviewers' comments:

Reviewer's Responses to Questions

**Comments to the Author**

1. Does this manuscript meet PLOS Global Public Health’s publication criteria? Is the manuscript technically sound, and do the data support the conclusions? The manuscript must describe methodologically and ethically rigorous research with conclusions that are appropriately drawn based on the data presented.

Reviewer #1: Yes

Reviewer #2: Yes

Reviewer #3: Partly

2. Has the statistical analysis been performed appropriately and rigorously?

Reviewer #1: Yes

Reviewer #2: Yes

Reviewer #3: Yes

3. Have the authors made all data underlying the findings in their manuscript fully available (please refer to the Data Availability Statement at the start of the manuscript PDF file)?

Reviewer #1: Yes

Reviewer #2: Yes

Reviewer #3: Yes

4. Is the manuscript presented in an intelligible fashion and written in standard English?

Reviewer #1: Yes

Reviewer #2: Yes

Reviewer #3: Yes

5. Review Comments to the Author

Reviewer #1: 1. From Lines 5 to 6, in the abstract section, "Data were analysed from a cohort study of child and adult residents (n=531)". The adult mentioned here, are they mother/child pairs or father/child pairs or both male and female adults?

2. In Line 10, on measured gastrointestinal illness, was water in the study areas assessed for heavy metal metals, microorganisms and permissible chemical standards?

3. In line 11, what is the provision for drawing blood samples from participants in terms of ethical issues relating to evasive procedure?

4. In Lines 19 and 20, belief systems, culture, and geographical locations are essential factors that must be considered. This recommendation should be location-specific.

5. From Lines 28 to 32, they seem to be contradictory. If the listed diseases are causes of morbidity and mortality in the target population, social and environmental issues can be causes of morbidity and mortality again. However, instead, they are determinants of health. This should be stated clearly.

6. In Line 34, it started with "Access to adequate housing, a human right and a core aim of Sustainable Development Goal 11 [4], is a promising yet neglected topic in SSA". However, from Lines 35 through 38, the statement started with "Features of inadequate housing, including indoor air pollution, overcrowding, unsafe water and poor sanitation . . .". There is a disconnect here. It would have been better if the following statement continued to outline features of adequate housing instead of inadequate housing.

7. In Line 72, Materials and Methods, the section did not outline detailed information on the study's methods.

8. In Line 76, are these two the only seasons in the country?

9. In Line 82, there was information on the treatment of malaria as part of the study design

10. in Line 96, on “Enrollment and follow up of study participants”, no detailed information was provided on the study participants and the number of study participants in the previous study. How many were lost to follow-up?

11. In Line 186, “Enrollment and follow up of study participants”, out of the total participants, how many children and how many were adults? Or is it mother-child pair?

12. From Lines 254 to 320 in the discussion section, the presentations are results and not discussion. Discussion started from Line 322.

13. In Line 374, how practicable can these recommendations be because the study area is rural. Most people are farmers who can barely afford basic daily needs much more, changing their housing patterns. Many other factors could result in gastrointestinal illness. What are the major factors responsible for this?

Reviewer #2: This manuscript does an excellent job of demonstrating the link between modern housing and three major causes of child mortality in Sub-Saharan Africa: malaria, ARI, and gastrointestinal illness. Some of the my initial concerns regarding the sampling and of residence and habitants was well documented.

Authors have provided information on the data for the study as well as important findings and necessary documentation.

The article has been written in an intelligible fashion and the choice of words are appropriate for easy understanding.

Reviewer #3: The topic is one of global public health interest as housing conditions play a role in health and disease. However, the manuscript has some methodological flaws that can affect the interpretation of the findings. Some factors that can contribute to acute respiratory infection and gastrointestinal illness in house occupants such as biomass fuel and indoor cooking where either broadly classified or not identified. The method of house selection was not described, and some human laboratory procedures were poorly described and not referenced. The authors robustly used descriptive and inferential statistics and presented their findings in clear tables. The manuscript is presented in standard English though there are some sentence structure errors that need to be corrected. The authors can improve on the readability of the manuscript by ensuring there is a logical progression in their writing. For instance ethical approval should be obtained before community entry.

In all, the topic carries interest, but the authors have not demonstrated novelty in their research, and the conclusion does not tie up with the objective of the research.

6. PLOS authors have the option to publish the peer review history of their article (what does this mean?). If published, this will include your full peer review and any attached files.

**Do you want your identity to be public for this peer review?** For information about this choice, including consent withdrawal, please see our Privacy Policy.

Reviewer #1: No

Reviewer #2: No

Reviewer #3: No

---

## [Editor Report · Decision Letter 1]

15 Nov 2021

House design and risk of malaria, acute respiratory infection and gastrointestinal illness in Uganda: a cohort study

PGPH-D-21-00703R1

Dear Dr. MUSIIME,

We're pleased to inform you that your manuscript has been judged scientifically suitable for publication and will be formally accepted for publication once it meets all outstanding technical requirements.

Within one week, you'll receive an e-mail detailing the required amendments. When these have been addressed, you'll receive a formal acceptance letter and your manuscript will be scheduled for publication.

An invoice for payment will follow shortly after the formal acceptance. To ensure an efficient process, please log into Editorial Manager at https://www.editorialmanager.com/pgph/ click the 'Update My Information' link at the top of the page, and double check that your user information is up-to-date. If you have any billing related questions, please contact our Author Billing department directly at authorbilling@plos.org.

Kind regards,

Nnodimele Onuigbo Atulomah, PhD

Academic Editor

Additional Editor Comments:

Kindly include the references to the ethical approvals provided in the appropriate narrative.